# Low-frequency theta oscillations in the human hippocampus during real-world and virtual navigation

Véronique D. Bohbot[1], Milagros S. Copara[2], Jean Gotman[3] & Arne D. Ekstrom[2,4,5]

Low-Frequency Oscillations (LFO) in the range of 7–9 Hz, or theta rhythm, has been recorded in rodents ambulating in the real world. However, intra-hippocampus EEG recordings during virtual navigation in humans have consistently reported LFO that appear to predominate around 3–4 Hz. Here we report clear evidence of 7–9 Hz rhythmicity in raw intra-hippocampus EEG traces during real as well as virtual movement. Oscillations typically occur at a lower frequency in virtual than real world navigation. This study highlights the possibility that human and rodent hippocampal EEG activity are not as different as previously reported and this difference may arise, in part, due to the lack of actual movement in previous human navigation studies, which were virtual.

[1] Douglas Institute, Department of Psychiatry, McGill University, 6875 boulevard LaSalle, Verdun, Quebec, Canada H4H 1R3. [2] Neuroscience Graduate Group, University of California, Davis, 1 Shields Avenue, Davis, California 95616, USA. [3] Montreal Neurological Institute, McGill University, 3801 University Street, Montreal, Quebec, Canada H3A 2B4. [4] Center for Neuroscience, University of California, Davis, 1 Shields Avenue, Davis, California 95618, USA. [5] Department of Psychology, University of California, Davis, 1 Shields Avenue, Davis, California 95618, USA. Correspondence and requests for materials should be addressed to V.D.B. (email: veronique.bohbot@mcgill.ca) or to A.D.E. (email: adekstrom@ucdavis.edu).

Rhythmic slow activity or the theta rhythm, is a neural pattern first observed in the electroencephalogram (EEG) of the rodent hippocampus during voluntary movement and associated with arousal, spatial memory and motor behaviour[1–3]. Constraining rodents to a running wheel[4] confirmed these findings: while their location remained constant, theta power increased with running speed[5]. Suggesting its importance to spatial memory, eliminating low-frequency oscillations (LFO) in the theta band impairs spatial learning and memory in rodents[3]. Studies in virtual reality (VR) show that humans also exhibit LFO across the hippocampus during virtual movement[6,7]. LFO, however, manifests at a lower frequency than those observed in rodents, peaking during virtual movements around 3.3 Hz in humans and 7.7 Hz in rats[8]. In rodents, movement in VR attenuates LFOs relative to real-world movement[9]. Similarly, another study suggested that virtual, as opposed to real-world movement, reduced LFO frequency and abolished the relationship between movement speed and LFO in rodents[10], suggesting the importance of motor, vestibular and proprioceptive input to LFO. Some have thus argued that VR, because it lacks such body-based input, makes a poor approximation of reality[11] while others have suggested that LFO is absent altogether in non-human primates[12] although LFO was observed during ocular movements in active scene viewing[13]. Another possibility is that VR and other memory-related processing induces oscillations at lower frequencies than real-world movement[10,14,15]. We explored these ideas by testing, for the first time, freely ambulating patients with depth electrodes, including three contacts per electrode, in their hippocampus. We also tested them during VR navigation, allowing us to compare the two conditions.

## Results

### Human hippocampus EEG recordings during real navigation.

Five patients (see Table 1) explored a real-world environment in a task modelled after the Morris Water Maze, by searching for five hidden locations in a room (Fig. 1a)[16]. Raw traces (Fig. 1b–d) during search for and recall of hidden locations revealed clear evidence of low-frequency rhythmicity. The power spectral density showed peaks around 7–9 Hz on some contacts, which were consistently significantly greater than stop periods in the same session (Fig. 2a). Interestingly, though, we found that LFOs were somewhat variable, even for the same task. We also found raw traces during real world searching showing a peak at lower frequencies, between 1–4 Hz (Figs 1c and 2b). We observed similar increases in LFO during recall of the previously learned targets (Figs 1d and 2c) and during real-world walking, in which patients walked unconstrained (Figs 1e and 2d, see Methods). We then tabulated the total number of hippocampal contacts showing significant differences between real-world navigation and stopping: real-world random walking > stop, recall > stop, search > stop and the opposite contrasts (for example, stop > search) within the 1–12 Hz band. Most strikingly, as shown in Table 2, many contacts showed significantly increased LFO during real-world walking compared with stop periods

across the 1–12 Hz band (Binomial test, random walking: 25 versus 0 unique significant contacts out of 30 hippocampal recordings, that is, $83 \pm 18\%$ versus $0 \pm 12\%$, $P < 0.0001$). Consistent with our observation of variability within task across the 1–12 Hz frequency band, the number of contacts showing more LFO during searching than stop periods within 1–4, 4–8 and 8–12 Hz bands did not differ from an even distribution across these subfrequencies (Chi Square goodness of fit test, $\chi^2(2) = 1.0$, $P = 0.6$, Table 2). Similarly, we found more unique contacts across the 1–12 Hz band showing LFO increases for recall versus stopping (Binomial tests, $63 \pm 19\%$ versus $10 \pm 17\%$, $P < 0.001$) and marginally significantly more for searching versus stopping ($40 \pm 17\%$ versus $10 \pm 17\%$, $P < 0.08$, Table 2). The number of significant contacts showing recall > stop and search > stop again did not differ significantly within the three bands ($\chi^2(2) = 2.0$, $P = 0.3$, $\chi(2)^2 = 3.4$, $P = 0.2$, Table 2), although, there were more contacts showing movement related effects in the theta and alpha bands (9 and 9 contacts) compared with the delta band (three contacts), see Table 2. Altogether, these findings suggest that (1) real-world movement elicits LFO (2) these are reduced during stopping periods (Fig. 1f).

### Human hippocampus EEG recordings during virtual navigation.

We next considered recordings when the same patients navigated an 8-arm virtual maze (Fig. 1g)[17]. We again found LFO (Figs 1h and 2e), although raw traces suggested that these often occurred in a lower frequency than during real-world navigation. Comparing the distribution of unique contacts for virtual navigation across the three bands (1–12 Hz; Table 2), we found a marginally significantly higher number of contacts showing higher LFO during VR navigation than during stop periods ($43 \pm 17\%$ versus $10 \pm 15\%$, Binomial test, $P < 0.06$; Fig. 1i). We again found that the distribution of contacts across the three bands during virtual navigation did not differ from an even distribution of contacts (Chi square goodness of fit test, $\chi^2(2) = 3.3$, $P < 0.2$, Table 2).

### Hippocampal EEG is different in real and virtual navigation.

We then compared the distribution of contacts during virtual navigation showing movement > stop effects with the same contrast for real-world searching, which were the closest analogues behaviourally (see Methods). Both conditions involved freely searching for specific locations repeatedly during navigation. Using a $2 \times 3$ Fisher's exact test to assay for associational differences, we found a significant cross over interaction (Table 2 row 1 versus row 7, $P < 0.04$, Fig. 3). Specifically, this effect appeared driven by a greater number of contacts showing increases in the delta band during virtual than real-world searching compared with the alpha band (delta: 10 versus 3 contacts and alpha: 4 versus 9 contacts, Table 2). This supported the hypothesis that virtual movement led to greater differences within lower frequency bands (delta) than real-world searching. Note, however, that we did observe some variability in frequency in relation to the task, with the real world condition showing significant contacts within the 1–4 Hz frequency

| Table 1 | Patient demographics. | | | | | |
|---|---|---|---|---|---|---|
| **Patient** | **Sex** | **Handedness** | **Age** | **Testing days after implant** | **Medication** | **Occupation** |
| #1 | Male | Right | 15 | 8 | Dilantin, Phenobarbital | Student |
| #2 | Male | Right | 20 | 18 | Tegratol | Student |
| #3 | Female | Right | 45 | 12 | Zoloft, Losec, Estragel | Waitress |
| #4 | Female | Right | 43 | 11 | Dilantin, Phenobarbital | Secretary |
| #5 | Female | Right | 52 | 6 | Neurontin | Homemaker |

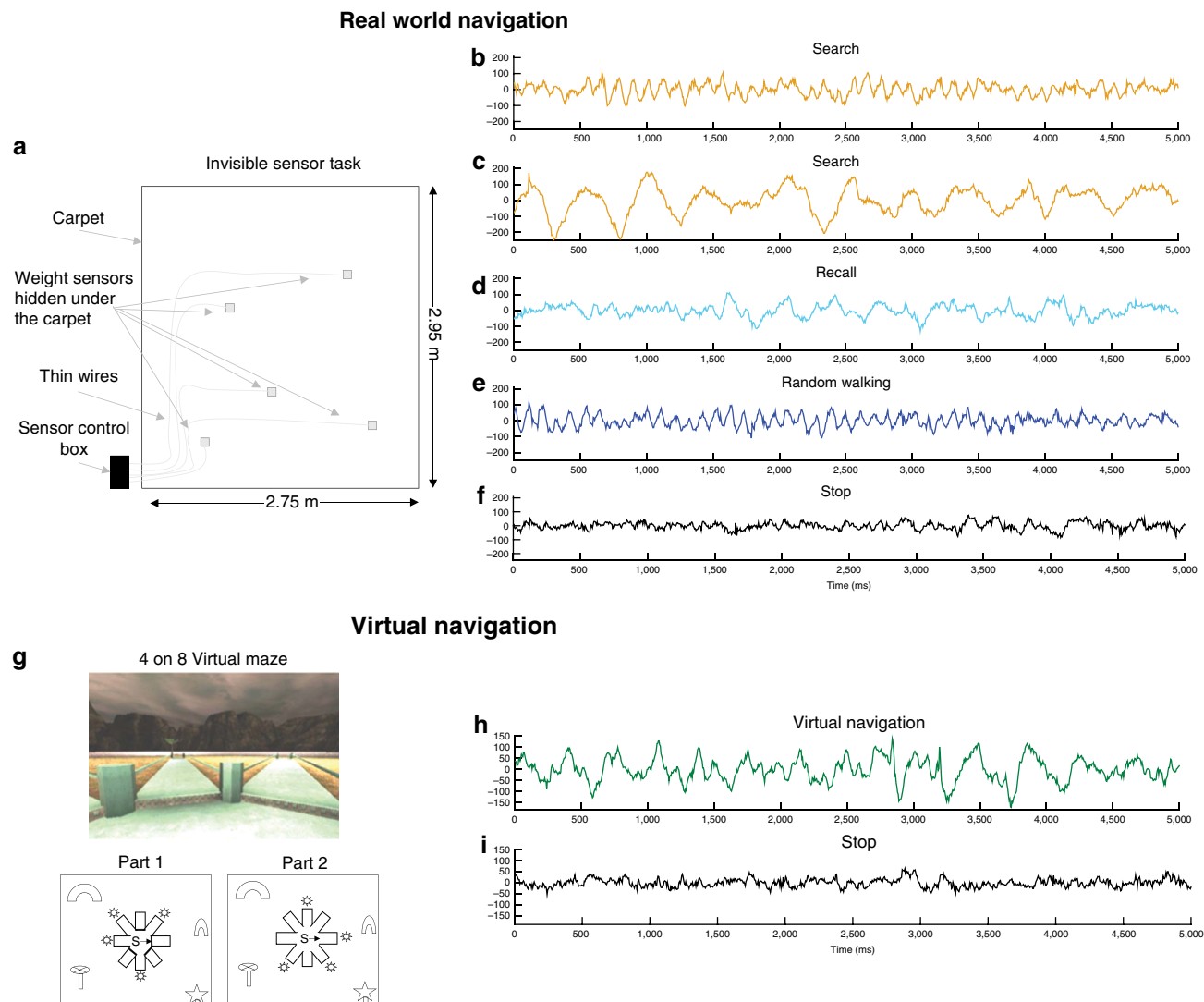

**Figure 1 | Overview of behavioural tasks and 5-s traces from hippocampal contacts during real-world conditions.** (**a**) Top view of the space patients navigated while searching for invisible sensors (thin square plate) underneath the carpet. (**g**, top) First person view of the virtual navigation task showing landmarks in the distance as well as views of the radial arms in the task. (**g**, bottom) Bird's eye view of radial arms layout. Patients were instructed to search for target objects (denoted by the suns) at the bottom of a pit located at the end of the arm. In part 1, patients could only enter four arms to pick up the objects. In part 2, patients had to remember previously visited arms and avoid them in order to retrieve the target objects in the new arms. (**b**) Posterior hippocampal contact from patient #2 demonstrates high-theta oscillations during real-world search for the invisible sensors. (**c**) Hippocampal contact from patient #4 demonstrates low-theta oscillations during real-world search for an invisible sensor. (**d**) Hippocampal contact from patient #5 demonstrates low to mid-range theta oscillations during recall of spatial locations in the invisible sensor task. (**e**) Posterior hippocampal contact from patient #2 demonstrates theta oscillations during real-world random walking. (**f**) Hippocampal contact from patient #2 shows minimal oscillations when patient is stopped and standing still. (**h**-**i**) Five-second raw traces from hippocampal contacts during the VR task. (**h**) Hippocampal contact from patient #5 demonstrates low to mid-range theta oscillations during VR navigation. (**i**) Hippocampal contact from patient #2 shows minimal oscillations when patient is not moving.

band and the VR condition showing significant contacts within the 8–12 Hz band (Table 3). Thus, our findings suggest that while real world versus VR tend to elicit differences in the predominance of oscillations within higher versus lower theta bands, both conditions result in changes across the 1–12 Hz band; larger data sets of freely ambulating patients would be ideal in order to better quantify this difference.

## Discussion

Thus, our results demonstrate, for the first time, that freely ambulating humans show increased LFO ranging from 1–12 Hz compared with a standing position. This was true across different types of real-world movements: when patients walked around the room, searched for a hidden location by trial and error, and recalled locations they had learned. Notably, we observed significant numbers of electrode contacts in the hippocampus showing higher LFO during searching, recall and walking, which did not differ across subfrequencies within the 1–12 Hz band. These findings suggest the importance of ambulation to induction of LFOs but also the importance of memory processing[18]. These findings align with past studies from rodents suggesting that real-world movement induces LFO throughout the 1–12 Hz band[2,5,19], but importantly, extend them to the human hippocampus.

We also analysed a separate session from each patient involving virtual navigation of an 8-arm radial maze on a computer screen. We observed electrode contacts showing

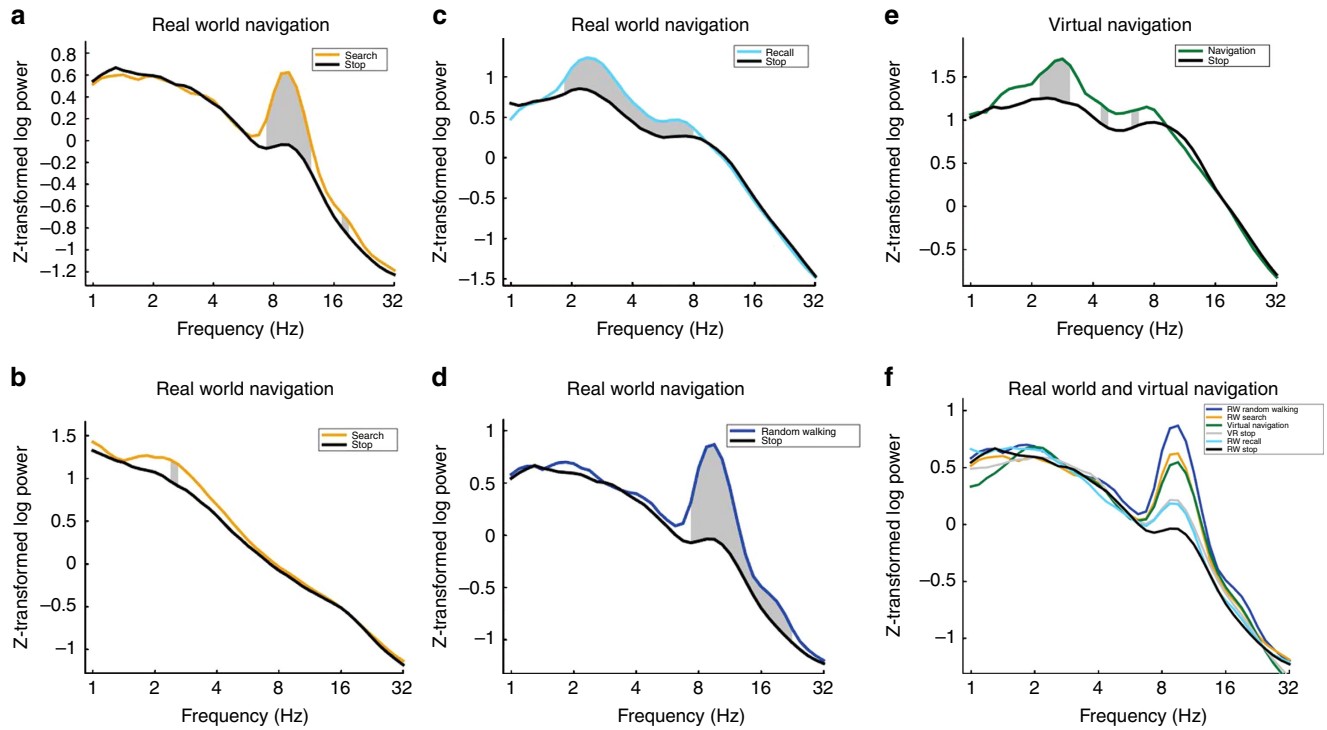

**Figure 2 | Individual power spectral density plots of experimental conditions versus stop periods within their respective real-world or virtual sessions.** Each plot is taken from a specific electrode contact in a patient (three patients are represented) that correspond to the raw traces in Fig. 1. Each condition's power spectral density differs from the contrasted condition by at least $P < 0.01$. (**a**) Real-world search (invisible sensor task) elicits more power in high-theta than when the patient is stopped (patient #2). (**b**) Real-world search (invisible sensor task) elicits more power in low-theta from patient #4 than when the patient is stopped. (**c**) Recall of real-world spatial locations elicited greater low-frequency oscillations (2–8 Hz) than the stop condition in patient #5. (**d**) Real-world random walk without cognitive demands elicits more power in high-theta than during stopping (hippocampal contact from patient #2). (**e**) In patient #5, navigation in a VR task (4-on-8 virtual maze) elicits greater low-theta oscillations than stopping in that session. (**f**) Comparisons across all experimental conditions within a single patient and electrode (patient #2). All conditions elicit low-frequency oscillations, real-world random walking eliciting the highest. Grey bar indicates a significant difference at $P < 0.01$.

**Table 2 | Electrode contacts showing significant differences between movement and stop periods.**

|  | 1–4 Hz | 4–8 Hz | 8–12 Hz | Unique 1–12 Hz |
|---|---|---|---|---|
| *Real world navigation* |  |  |  |  |
| RW search > stop | **3** | **9** | **9** | **12** |
| Stop > RW search | 3 | 0 | 0 | 3 |
| RW random walking > stop | 15 | 21 | 18 | 25 |
| Stop > RW random walking | 0 | 0 | 0 | 0 |
| RW recall > stop | 14 | 9 | 8 | 19 |
| Stop > RW recall | 2 | 0 | 1 | 3 |
|  |  |  |  |  |
| *Virtual navigation* |  |  |  |  |
| VR navigation > stop | **10** | **5** | **4** | **13** |
| Stop > VR navigation | 1 | 4 | 5 | 3 |

Each column depicts a canonical frequency band (delta: 1–4 Hz, theta: 4–8 Hz, alpha: 8–12 Hz) and the respective number of electrodes showing significant differences between movement and stop conditions. For example, in the Real world search versus stop condition, three electrode contacts showed significant differences between conditions within frequencies 1–4 Hz, evidenced by a *t*-test ($P < 0.01$). These counts are separated into contrasts comparing real-world navigation and virtual navigation. Because some contacts showed significant effects in multiple frequency bands (for example, 1–4 Hz and 4–8 Hz), the final column ('unique 1–12 Hz') tabulates the total number of unique contacts showing effects across the 1–12 Hz band for each comparison. The *P* values are indicated in bold.

increases in LFO during virtual movement (that is, keyboard pressing-induced visual optic flow) compared with stop periods. We noted, however, that these effects typically manifested within lower frequency bands than LFO during real-world movement. Specifically, we found a significant crossover interaction effect such that virtual navigation induced lower-frequency activity than searching for a hidden sensor in the real world. These findings are consistent with human studies suggesting that virtual navigation tends to result in low-frequency hippocampal oscillations peaking around 3.3 Hz (refs 8,14,20). Real-world navigation in rodents, on the other hand, typically generates LFO at a higher frequency between 8–12 Hz (refs 2,8,9).

Altogether, our findings help resolve an important debate regarding LFO in the human hippocampus. First, we show that prominent LFOs are present during real-world searching, walking, and recalling locations, and that these effects are present across the low-frequency spectrum (1–12 Hz). These effects were consistently higher during movement than stopping periods, implicating movement itself as one generating mechanism for these oscillations. Second, we found that virtual movement induced oscillations peaking within a lower frequency band than real-world searching, as indicated by a significant crossover interaction. These findings confirm previous theoretical models in humans suggesting that virtual movement may shift the frequency-wise prevalence to a lower range than real-world movement[14], possibly reflecting missing motor, vestibular and proprioceptive input[15]. These results illustrate the possibility that human hippocampal EEG activity during real-world movement is more similar to rodent EEG activity than previously reported.

## Methods

**Patients.** We studied five patients investigated for surgical treatment of their medically refractory epilepsy (2 men, ages 15 and 20; 3 women, ages 43, 45 and 52—see Table 1). Informed consent was obtained by all research participants. All

research was conducted in accordance to institution guidelines of the Research Ethics Board of the Montreal Neurological Institute. The pre-implantation hypothesis regarding the epileptic generator involved frontal and temporal regions in three patients, parietal and temporal regions in one, and temporal regions in one. One patient (#2) has two implanted electrodes in each hippocampus for a total of four electrodes, two patients (#3 and #4) had bilateral temporal implantations of a single electrode per hippocampus, and two patients had a temporal implantation of a single electrode unilaterally (#1 and #5). Electrodes were manufactured on site (nine contacts, 0.5–1 mm in length and separated by 5 mm). Electrode implantation was performed with frameless stereotaxy according to the method of Olivier et al.[21] All electrodes were implanted with entry orthogonal to the midline plane. Patients' antiepileptic medication was gradually reduced after implantation in an effort to facilitate seizure occurrence. Patients tolerated well the implantation and were usually feeling well and able to walk normally within 24 h of implantation. Patients participated in the real and virtual navigation studies on the same day, 6 to 18 days after intracranial surgical implantation of the electrodes (see Table 1). We only considered electrodes aimed at the hippocampus and retained only the first three contacts on each electrode, those localized in the hippocampus. We analysed EEG from a total of 10 hippocampal electrodes, that is, 30 contacts located in the hippocampus. We excluded periods of interictal spiking.

**Navigation tasks.** Patients participated in both a real-world navigation task and a virtual navigation task on a desktop computer.

*Real world navigation.* The real world navigation task, called the Invisible Sensor Task is an allocentric spatial memory task, that is, a task that requires navigation based on the relationship between environmental landmarks, where a target can be reached in a straight path, irrespective of the position of the observer. It was modelled after the Morris Water Task in rodents (see Bohbot et al.[16]). Participants learn the location of a target using distal landmarks present in the testing room. This task was previously shown to require the critical contribution of the parahippocampal cortex when participants searched for one target location after a delay[16]. However, this task requires the critical contribution of the hippocampus when participants have to rely on context or episodic memory to learn more than one sensor location[22]. In the current experiment, participants searched for five locations, one at a time, that could be identified by a tone when a thin sensor was triggered by the weight of the participant, that is, when stepped on. The weight sensors and wires were under a carpet and could not be identified either by sight or by a physical sensation under the foot.

*Search.* In the first trial, participants search for the invisible sensor by going back and forth, step by step, looking for the target using a systematic search taking care to remember and avoid previously visited regions of the carpet (avg. duration: 137.5 s) (ref. 23). After the invisible sensor was found, patients were instructed to make it beep three times with their foot, and look around in order to learn its position. Sensors were controlled by a control box with flip switches such that only one sensor was activated at a given time during this phase.

*Recall.* Four additional trials followed the search trial, whereby patients had to find the target location in a direct path, by starting the trial by the edge of the carpet, in the midline, facing the wall and back to the carpet (avg. duration: 14.67 s). The start positions were semi-randomized such that they were balanced across the four different sides of the carpet for each invisible sensory location. Hence, patients were given a total of five trials to learn the location of one sensor, before moving on to learning the location of the next sensor. The same procedure was repeated until the locations of all five sensors were learned. These could be recalled in any order.

*Walk.* All sensors were turned off and patients were instructed to walk around for ∼2 min (avg. duration: 112.4 s, epoched into 20 s bins).

*Stop.* Recordings were made while patients stayed immobile and were interspersed among Recall periods (avg. duration: 30.75 s).

*Virtual navigation.* The 4-on-8 virtual maze is a navigation task modelled after the rodent radial maze, which was programmed with a commercially available computer game (Unreal; Epic Games, Raleigh, NC). The virtual task was presented

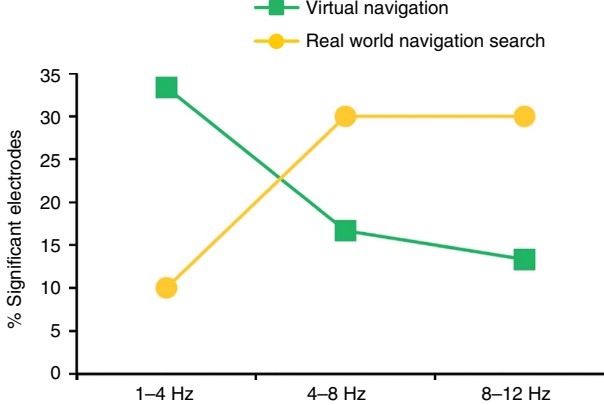

**Figure 3 | Electrode contacts showing significant differences between movement and stop periods.** Significant electrode contacts are depicted as the percentage of total electrode contacts across the established frequency bands.

**Table 3 | Number of electrodes showing statistically significant LFOs.**

| | Number of significant electrodes for the specified comparison (P < 0.01) | | | | | | | | | |
| --- | --- | --- | --- | --- | --- | --- | --- | --- | --- | --- |
| | **Experimental** | | | | | **Control** | | | | |
| Total no. of electrodes for each patient | | | | | | No. of significant electrodes for that comparison | | | | |
| | Search > stop | Walk > stop | Virtual > stop | Recall > stop | Total | Stop > search | Stop > walk | Stop > virtual | Stop > recall | Total |
| *1–4 Hz* | | | | | | | | | | |
| Patient #1 | 0 | 0 | 3 | 0 | **3** | 0 | 0 | 0 | 0 | **0** |
| Patient #2 | 0 | 5 | 4 | 5 | **14** | 0 | 0 | 1 | 0 | **1** |
| Patient #3 | 0 | 3 | 0 | 2 | **5** | 3 | 0 | 0 | 0 | **3** |
| Patient #4 | 3 | 6 | 2 | 5 | **16** | 0 | 0 | 0 | 0 | **0** |
| Patient #5 | 0 | 1 | 1 | 2 | **4** | 0 | 0 | 0 | 2 | **2** |
| Total | **3** | **15** | **10** | **14** | 42 | **3** | **0** | **1** | **2** | 6 |
| *4–8 Hz* | | | | | | | | | | |
| Patient #1 | 1 | 3 | 1 | 2 | **7** | 0 | 0 | 0 | 0 | **0** |
| Patient #2 | 5 | 9 | 1 | 1 | **16** | 0 | 0 | 3 | 0 | **3** |
| Patient #3 | 0 | 1 | 0 | 1 | **2** | 0 | 0 | 0 | 0 | **0** |
| Patient #4 | 3 | 6 | 1 | 3 | **13** | 0 | 0 | 1 | 0 | **1** |
| Patient #5 | 0 | 2 | 2 | 2 | **6** | 0 | 0 | 0 | 0 | **0** |
| Total | **9** | **21** | **5** | **9** | 44 | **0** | **0** | **4** | **0** | 4 |
| *8–12 Hz* | | | | | | | | | | |
| Patient #1 | 0 | 1 | 1 | 1 | **3** | 0 | 0 | 0 | 0 | **0** |
| Patient #2 | 5 | 10 | 2 | 2 | **19** | 0 | 0 | 2 | 0 | **2** |
| Patient #3 | 1 | 0 | 0 | 2 | **3** | 0 | 0 | 1 | 0 | **1** |
| Patient #4 | 3 | 5 | 1 | 3 | **12** | 0 | 0 | 2 | 0 | **2** |
| Patient #5 | 0 | 2 | 0 | 0 | **2** | 0 | 0 | 0 | 1 | **1** |
| Total | **9** | **18** | **4** | **8** | 39 | **0** | **0** | **5** | **1** | 6 |

The P values are indicated in bold.

to patients on a computer screen and they used arrow keys to move in the virtual environment. Patients always start in the center of the maze, facing the same direction. The maze is surrounded by distal landmarks such as a sunset and mountains, as well as proximal landmarks, such a two distinct trees. In a first part, 4 out of 8 arms are closed and patients are instructed to enter the open arms in order to retrieve objects. Since objects are retrieved by going down stairs leading to a pit, at the very end of the arm, patients can not see objects from the choice point. Therefore, patients have to remember where they picked up the objects in the first part, in order to avoid going to the same places in the second part of that trial. In part two of a trial, all eight arms are made accessible to participants and they have to avoid visiting previously entered pathways in order to find the remaining four target objects. Participants were given between two and three trials[17,22]. This task has previously been shown to be sensitive to functional magnetic resonance imaging (fMRI) activity and grey matter in the hippocampus and caudate nucleus of healthy adults[17,24].

*Stop.* Recordings were made while patients stayed immobile (avg. duration: 45.2 s).

The real-world search condition was selected for comparison with the virtual navigation epochs because it best represented the period during which patients had to search for the sensor to learn its location in the environment. During the real-world search, patients had to remember the areas of the environment that were previously visited in order to avoid these areas until the target is found. Similarly, in the virtual environment, patients search for target objects and in a second phase, they have to remember the areas of the environment that were previously visited in order to avoid these areas until the target is found. For this reason, we believe that the two conditions are the most closely matched and result in a valid comparison.

**Analysis.** The EEG signal was high pass-filtered at 0.3 Hz, low-pass filtered at 70 Hz and sampled at 200 Hz. EEG were recorded using the Harmonie EEG system (Stellate, Montreal, Qc, Canada). Movement artifacts were manually removed by visually inspecting raw EEG traces on the Harmonie system. These were then ported into MATLAB for post-processing.

Spectral power estimates were computed by convolving the filtered signal with six cycle Morlet wavelets at 32 logarithmically spaced frequencies ranging from 1 to 45 Hz. Because our hypotheses are concerning LFOs we focused on frequencies lower than 32 Hz. These power values were binned into delta (1–4 Hz), theta (4–8 Hz) and alpha (8–12 Hz) frequency bands. Power values were also subsequently log transformed and then z-transformed. The number of electrode contacts with significantly different power values across experimental conditions was determined with $t$-tests across each frequency band at a $P$ value < 0.01. Thus, we averaged the log and z-transformed power for delta (1–4 Hz), theta (4–8 Hz), and alpha (8–12 Hz) bands for each condition (for example, Search). The data were epoched with respect to the natural occurrence of the events during the tasks (see average event duration in task description), except for Walk which was one long event that we parsed into 20 s events. We then compared these values using a $t$-test between conditions (for example, search versus stop period). We corrected for multiple comparisons by bootstrapping the power values for each trial between movement conditions and stop conditions, conducting a one tailed $t$-test at each frequency. We extracted individual $t$-distributions for each patient as there were different numbers of trials for certain conditions, giving us variable degrees of freedom for each patient. When bootstrapped, each patient's $P$ value that separated the top 5% of $t$-values was approximately $P < 0.01$. We applied an alpha value of $P < 0.01$ (two-tailed) to conservatively correct for multiple comparisons across subfrequencies (delta, theta, alpha). For all movement conditions, all counts well exceeded those predicted by chance (that is, 1%, or 1 electrode contact, see Table 3).

To compare different conditions (for example, search versus stop), we adopted two different approaches. Consistent with past work[25], we performed binomial tests, which estimate the confidence intervals for a significant number of electrode contacts compared with a chance distribution. We then compared the confidence intervals between different conditions, which allowed us to assess for significant differences between conditions. Additionally, we performed Chi Square goodness of fit tests on electrode counts, consistent with past work[6]. We did this by taking the electrode contacts across a frequency band and comparing with an even distribution of electrode contacts across these three frequency bands. Finally, we performed Fisher's exact tests on electrode counts across frequency bands. This allowed us to look for crossover interaction effects, in other words, a differential distribution of electrodes as a function of both condition and frequency band.

**Data availability.** The data sets generated during and/or analysed during the current study are available from the corresponding author on reasonable request.

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

## Acknowledgements

We are grateful to Nicolas Andersen for his help with the manual removal of the motion artifacts. This work was supported by CIHR Grant no. 86727 and 301763, NSERC grant no. 239896, NIH/NINDSR01NS076856 and NSF BCS-1630296.

## Author contributions

Data were collected at the Montreal Neurological Institute between the years 1999 to 2000 by V.D.B. and J.G. Data analyses were carried out by M.S.C., A.D.E. and V.D.B. The manuscript was written by V.D.B., A.D.E., M.S.C. and J.G.

## Additional information

**Publisher's note**: 

