## [Peer Review File · Nature Communications]

Reviewers' Comments:

Reviewer #1 (Remarks to the Author)

Movement-related theta rhythm in the hippocampus is a robust and dominant feature of the local field potential of experimental animals such as rats and mice. Attempts to understand the computational and information-processing significance of this rhythm have led to numerous fundamental discoveries and insights into hippocampal function (in particular) and neural system dynamics (in general). However, it has been difficult to relate these findings to human hippocampus due to a longstanding controversy about whether humans (and primates in general) show the same type of theta under the same conditions as rodents. This brief communication provides compelling evidence that human hippocampus can display movement related theta, of the same frequency and under the same conditions, as rodents. Although there are some concerns with the data, allowances must be made due to the difficulty in obtaining these data from freely moving patients. I find the overall pattern of results to be convincing and important. These results will help bridge the gap between human and rodent work, assisting the effort to apply the principles learned from rodent work (concentrating on spatial maps and place/grid cells) to understanding the role of the human hippocampus in declarative memory.

- 1) Lines 70-72. Where do the numbers 83% and 10% come from? I do not understand how these numbers are calculated from the data in Figure 3A. Please clarify here and throughout (line 76, 88).
- 2) Lines 91-92. It is debatable which of the real world conditions most closely matches the virtual navigation condition. The authors should justify this better. The comparison they make has the feel of cherry-picking, in that they chose the real-world condition to compare against the virtual condition based on the pattern of data after the fact and the barely significant result. This is my greatest concern about the paper that the analysis of Figure 3B was not a principled one but instead was a post hoc decision.
- 3) Why does Figure 3A not show results from Stop > RW Recall?

Reviewer #2 (Remarks to the Author)

Summary

In their manuscript, Bohbot et al. present results from a human electrophysiology study of real and virtual navigation. Whereas numerous studies, including many by one of the corresponding authors, have explored the electrophysiological correlates of virtual navigation in humans with recordings from depth electrodes in the hippocampus and other medial temporal lobe regions, the authors claim this is the first to study both virtual and real navigation (where the participant

actually is in motion around a room performing a cognitive task) in humans. They found that real-world navigation does induce oscillations (or increased amplitudes) in the hippocampus that are similar to those observed in rats performing locomotion. Interestingly, these oscillations occurred at a higher spectral frequency than the increase in power observed during virtual navigation.

These results are important for a number of reasons. First, it is important to show that real-world navigation induces oscillations in humans in a similar way to rodents, helping to unify the two domains. Second, just looking within humans, it is interesting to see the increased frequency (around 7-9Hz instead of 1-4Hz) for real-world vs. virtual navigation. The results for virtual reality replicate many previous results in the literature, which themselves have spawned much debate as to why the frequencies observed in human virtual navigation are lower than the frequencies observed in rodent navigation. This study highlights the possibility that the difference in hippocampal activity between humans and rodents is not as large as previously reported and this difference is likely due to the lack of actual movement in human virtual navigation. Finally, the real-world navigation task is innovative and is directly analogous to a Morris water maze task, which has been used quite often in rodent studies of memory-guided navigation.

That said, as I outline below, there are a number of points that require clarification, including both the narrative and the statistical approaches.

Major Points

- I think the issues and importance of the results that I outline in the summary above should be better spelled out in the manuscript, itself. The intro as written is fragmented (it reads like a bullet list of semi-related one-line summaries with citations) and does not properly motivate the study or focus the reader on the importance of the results.
- Are the significant regions in grey presented within-subjects in Figure 2 corrected for multiple comparisons? I can see from the supplemental methods that the main analyses are corrected for the three frequency bands, but I don't see how the grey regions were determined.
- Why are counts of significant electrodes the proper way to assess significance? On a related note, how do you calculate the number of electrodes that would be predicted by chance (1 electrode)? Also, please motivate the use of the Chi Square and Fisher's exact tests.
- Were the statistical comparisons performed once across all subjects or within subject and combined across subjects?
- Since the part of the story is about Real vs Virtual Navigation and all participants performed both tasks, shouldn't the comparisons be made within subject and combined across subject, not just as a sum of electrodes? Is the issue that there are only five participants? I think my confusion is arising from a lack of clarity in the methods.

Minor Points

- Can you better characterize the movement-related artifacts in the data during the real-world data and how you removed them?

Reviewer #3 (Remarks to the Author)

The paper by Bohbot et al. has been investigating oscillatory activity in the hippocampus of freely moving patients. As I understand the main finding is that clear 7-9 Hz theta activity emerges in at least some patients when they are actively exploring. A lower effect at 1-4 Hz was observed when (some) patients were navigating in a virtual maze or doing memory-related tasks but not moving. This is a highly important finding since it resolves a debate on the frequency of the human hippocampal theta. Previous reports have reported the human hippocampal theta to be in the 1-4 Hz range; however, this finding now demonstrates that the real-world navigation related theta in humans can be close the exploration related theta in rats. I do however lack a clear summary in the text of what findings can be considered robust (while the opening of the paper is quite clear about the 7-9 Hz theta, the concluding paragraphs in line 99 and line 119 only mentions the 1-12 Hz band). I feel that these exciting findings suffers from a somewhat unclear presentation. Nevertheless, I anticipate that this study will spark future MTL recordings in humans during real-world explorations.

Major concerns:

- The paper was a difficult read despite a conceptual simple study. The presentation does feel somewhat 'cherry picked' and I would encourage the authors to do a more complete data presentation. I'm aware of the practical complications acquiring these data and differences in subjects due to the placement of the electrodes etc. I also do realize that the purpose of the paper is to provide proof-of-principle for real-world navigation related high-frequency theta.
- I realize that there was quite some variability between subjects in terms of frequency in relation to task. I propose to be explicit on this and articulate the need to collect more data in patients during real-world navigation to get a complete picture.
- In Figure 2, different conditions are presented for different subjects. Why not show all the conditions per subject?
- The main text should make clear how many patients were used? How many contacts in hippocampus? How was the hippocampus identified given that the electrodes also go in rhinal cortex?
- Why use a logarithmic scale in the spectra Fig 2? It makes the 7-9 Hz theta peak looks narrower but I would prefer a linear scale as more typically use for low-frequency spectra.
- The spectra points to 7-9 Hz ; later 4-8 Hz and 8-12 Hz are quantified. This is confusing.
- Figure 3A; table hard to understand. Explain in caption.

Reviewer #1 (Remarks to the Author)

Movement-related theta rhythm in the hippocampus is a robust and dominant feature of the local field potential of experimental animals such as rats and mice. Attempts to understand the computational and information-processing significance of this rhythm have led to numerous fundamental discoveries and insights into hippocampal function (in particular) and neural system dynamics (in general). However, it has been difficult to relate these findings to human hippocampus due to a longstanding controversy about whether humans (and primates in general) show the same type of theta under the same conditions as rodents. This brief communication provides compelling evidence that human hippocampus can display movement related theta, of the same frequency and under the same conditions, as rodents. Although there are some concerns with the data, allowances must be made due to the difficulty in obtaining these data from freely moving patients. I find the overall pattern of results to be convincing and important. These results will help bridge the gap between human and rodent work, assisting the effort to apply the principles learned from rodent work (concentrating on spatial maps and place/grid cells) to understanding the role of the human hippocampus in declarative memory.

R1.1) Lines 70-72. Where do the numbers 83% and 10% come from? I do not understand how these numbers are calculated from the data in Figure 3A. Please clarify here and throughout (line 76, 88).

We thank the reviewer for his/her positive and constructive comments on our manuscript. We appreciate the opportunity to respond to clarifications about our approach and methods. The significance counts (in this case, the numerator for our percentages above) came from considering the total number of **unique** electrodes within the 1-12 Hz frequency band that showed movement > stopping effects and stopping > movement effects in a two-sample one-tailed t-tests (to provide directionality) corrected at $p < .01$. The total number of hippocampal electrodes across all patients was 30 (in this case, the denominator for the percentages). Thus, in the case of 83%, we had 25 significant electrodes contacts and in the case of 10%, 3 significant electrode contacts. We have attempted to clarify this issue throughout the manuscript.

R1.2) Lines 91-92. It is debatable which of the real world conditions most closely matches the virtual navigation condition. The authors should justify this better. The comparison they make has the feel of cherry-picking, in that they chose the real-world condition to compare against the virtual condition based on the pattern of data after the fact and the barely significant result. This is my greatest concern about the paper that the analysis of Figure 3B was not a principled one but instead was a post hoc decision.

We appreciate this concern and agree that we need to provide better justification for the rationale behind this comparison. The real world search condition was selected because it best represented the period during which patients had to search for the sensor to learn its location in the environment. During the real world search, patients had to remember the areas of the environment that were previously visited in order to

avoid these areas until the target is found. Similarly, in the virtual environment, patients search for target objects and in a second phase, they have to remember the areas of the environment that were previously visited in order to avoid these areas until the target is found. For this reason, we believe that the two conditions are the most closely matched and result in a valid comparison. This information was added to the Supplementary Methods.

R1.3) Why does Figure 3A not show results from Stop > RW Recall?

We thank the reviewer for this clarification. The Stop > real world (RW) Recall was added to Figure 3A

Reviewer #2 (Remarks to the Author)

R2.1) "These results are important for a number of reasons. First, it is important to show that real-world navigation induces oscillations in humans in a similar way to rodents, helping to unify the two domains. Second, just looking within humans, it is interesting to see the increased frequency (around 7-9Hz instead of 1-4Hz) for real-world vs. virtual navigation. The results for virtual reality replicate many previous results in the literature, which themselves have spawned much debate as to why the frequencies observed in human virtual navigation are lower than the frequencies observed in rodent navigation. This study highlights the possibility that the difference in hippocampal activity between humans and rodents is not as large as previously reported and this difference is likely due to the lack of actual movement in human virtual navigation. Finally, the real-world navigation task is innovative and is directly analogous to a Morris water maze task, which has been used quite often in rodent studies of memory-guided navigation."

"- Are the significant regions in grey presented within-subjects in Figure 2 corrected for multiple comparisons? I can see from the supplemental methods that the main analyses are corrected for the three frequency bands, but I don't see how the grey regions were determined."

We appreciate the reviewer's constructive and positive comments on our manuscript. We also appreciate the chance to explain how we corrected for multiple comparisons in the power spectral density (psd) plots. Briefly, we did this through a non-parametric permutation approach. Specifically, consistent with past work (e.g., Watrous et al. 2011 J Neurophysiology), we bootstrapped the power values for each trial between movement conditions and stop conditions, conducting a one tailed t-test at each frequency. We extracted individual t-distributions for each patient as there were different numbers of trials for certain conditions, giving us variable degrees of freedom for each patient. The results from these permutations, considering p values across the different patients, indicated that an appropriate p value to control for multiple comparisons was $p < .01$. We have added this information to the Supplementary Methods.

R2.2) "Why are counts of significant electrodes the proper way to assess significance? On a related note, how do you calculate the number of electrodes that would be

predicted by chance (1 electrode)?

We appreciate the opportunity to address this important issue regarding our methodological approach in this study. Considering significance at each individual electrode (or cell) is generally considered the “gold” standard for statistical analyses in electrophysiology, particularly when analyzing the LFP and when sufficiently large numbers of samples are collected. This approach is also consistent with the vast majority of past approaches to large-sample LFP and cellular recording studies, particularly in humans but also in monkeys (e.g., Sederberg et al. 2003 J Neuro; Rolls and Xiang 2005 J Neuro; Ekstrom et al. 2005 Hippocampus; Watrous et al. 2011 J Neurophys). This approach is preferable to using grand averages, which is more common in the scalp EEG literature, and which uses a standardized electrode placement procedure. This is because intracranial electrodes are placed solely based on clinical determinations, and electrode contacts that happen to be closer to large sources will pick up stronger signal, which will then dominate in the grand average. Thus, considering each electrode individually thus allowed us to better deal with variability in individual electrode placement, while at the same time is consistent with most past approaches to invasive recordings that employ sufficiently large samples for Chi square tests to be meaningful.

R2.3) Also, please motivate the use of the Chi Square and Fisher's exact tests.”

We employed Chi Square goodness of fit tests to compare the distribution of electrodes across frequency bands to a null distribution in which the counts did not differ across frequency bands. This is a standard application of the Chi square test (e.g., Kreysig 1993 Advanced Engineering Mathematics) and is consistent with numerous past ECoG and single neuron recording studies in humans (e.g., (e.g., Sederberg et al. 2003 J Neuro; Rolls and Xiang 2005 J Neuro; Ekstrom et al. 2005 Hippocampus; Watrous et al. 2011 J Neurophys). In most cases, we did not see a deviation from an even distribution of electrodes across frequency bands. However, these comparisons did not involve a direct comparison of electrode differences *between* conditions (e.g., real vs. virtual) which is why we employed the Fischer's exact test. In contrast to the Chi square goodness of fit, the Fischer's exact test is intended instead to detect *associations* among categorical data sets, typically in 2x2 or 2x3 categorical situations. In our case, we wished to test whether there was a greater tendency for real world movement to be associated with higher frequency bands and virtual movement to be associated with lower frequency bands. Because we had counts of significance for each of these categories, (1-4 Hz, 4-8 Hz, 8-12 Hz), it was then natural to compare whether there were differential association (i.e., a dissociation). Our statistical comparison indicated that this was in fact the case.

R2.4) Were the statistical comparisons performed once across all subjects or within subject and combined across subjects?

All statistical comparisons were performed within electrode within patient, consistent with the majority of approaches to human LFP (please see R2.2). These were then tabulated across electrodes and patients, which we now detail in Supplemental Table 2.

R2.5) Since the part of the story is about Real vs Virtual Navigation and all participants performed both tasks, shouldn't the comparisons be made within subject and combined across subject, not just as a sum of electrodes? Is the issue that there are only five participants? I think my confusion is arising from a lack of clarity in the methods.

We appreciate this concern. The finding that movement shows greater power than stopping periods, first demonstrated by Vanderwolf in the 60s and widely replicated in both rats and humans, concerns the basic contrast of movement-related theta power being greater than that during still periods. In our paradigm, movement periods varied by condition (e.g., real world searching and virtual moving), as did stopping periods (e.g., stopping during real-world searching for targets vs. stopping in virtual reality). Because all movement > stop contrasts necessarily involved comparison first within a condition, we thus first performed these contrasts and then compared between the different experimental manipulations. The issue with directly comparing real world vs. virtual movement is that it would not take into consideration the corresponding stop periods, when we expect theta to be lower. We have tried to clarify these issues in the methods and provided a new table (Suppl Table 2) that provides more detail on the number of electrode contacts for each patient, as requested by the reviewer. This makes it clearer in terms of which patients show which effects. Overall, the table makes it clear that our effects at the group level that we originally reported are also present at the individual patient level.

R2.6) Can you better characterize the movement-related artifacts in the data during the real-world data and how you removed them?

Large amplitude movement related artifacts were removed manually with the software Harmonie after these were identified by visual inspection. The word “manually” was added to the supplementary methods in the sentence: “Movement artifacts were manually removed by visually inspecting raw EEG traces on the Harmonie system.”

Reviewer #3 (Remarks to the Author)

R3.1) The paper by Bohbot et al. has been investigating oscillatory activity in the hippocampus of freely moving patients. As I understand the main finding is that clear 7-9 Hz theta activity emerges in at least some patients when they are actively exploring. A lower effect at 1-4 Hz was observed when (some) patients were navigating in a virtual maze or doing memory-related tasks but not moving. This is a highly important finding since it resolves a debate on the frequency of the human hippocampal theta. Previous reports have reported the human hippocampal theta to be in the 1-4 Hz range; however, this finding now demonstrates that the real-world navigation related theta in humans can be close the exploration related theta in rats. I do however lack a clear summary in the text of what findings can be considered robust (while the opening of the paper is quite clear about the 7-9 Hz theta, the concluding paragraphs in line 99 and line 119 only mentions the 1-12 Hz band). I feel that these exciting findings suffers from a somewhat unclear presentation. Nevertheless, I

anticipate that this study will spark future MTL recordings in humans during real-world

-The paper was a difficult read despite a conceptual simple study. The presentation does feel somewhat 'cherry picked' and I would encourage the authors to do a more complete data presentation. I'm aware of the practical complications acquiring these data and differences in subjects due to the placement of the electrodes etc. I also do realize that the purpose of the paper is to provide proof-of-principle for real-world navigation related high-frequency theta.

We appreciate the reviewer's constructive comments on the manuscript. We have now provided a more complete data presentation in the Supplementary Methods and apologize for any unintended lack of clarity in our original presentation.

R3.2) I realize that there was quite some variability between subjects in terms of frequency in relation to task. I propose to be explicit on this and articulate the need to collect more data in patients during real-world navigation to get a complete picture.

We agree and have added the following to the manuscript: "Note, however, that we did observe some variability in frequency in relation to the task, with the real world condition showing significant contacts with the 1-4 Hz frequency band while the VR condition showed significant contacts within the 8-12 Hz band (Suppl. Table 2). Thus, our findings suggest that while real world vs. VR tend to elicit differences in the predominance of oscillations within higher vs. lower theta bands, both conditions result in changes across the 1-12 Hz band; larger data sets of freely ambulating patients would be ideal in order to better quantify this difference.

R3.3) - In Figure 2, different conditions are presented for different subjects. Why not show all the conditions per subject?

A complete data presentation for all participants is now available in the Supplementary Methods.

R3.4) - The main text should make clear how many patients were used? How many contacts in hippocampus?

This information was added to the main text in the last sentence of the second paragraph and in the first sentence of the third paragraph.

R3.5) How was the hippocampus identified given that the electrodes also go in rhinal cortex?

The target of the stereotaxic electrode placement is the hippocampus however, this was not confirmed with MRI in any of the patients. Our stereotaxic procedure, however, has yielded correct target placement of the electrodes based on post-implant electrode tracts, in other patients who received MRI after electrode removal. Patients in the current study, however, were tested before the MRI procedure became standard

practice. As such it is not possible to exclude the possibility that electrode contacts may have been in the entorhinal cortex with certainty.

R3.6) Why use a logarithmic scale in the spectra Fig 2? It makes the 7-9 Hz theta peak looks narrower but I would prefer a linear scale as more typically use for low-frequency spectra.

As can be seen in the plots below, both are valid options that look just as good. We favor the log scale.

REAL WORLD NAVIGATION

R3.7) The spectra points to 7-9 Hz ; later 4-8 Hz and 8-12 Hz are quantified. This is confusing.

To reduce the problem of multiple comparisons and to recognize the canonical frequency bands investigated in other studies, we restricted our quantification of significant electrode contacts to these windows: Delta (1-4 Hz), Theta (4-8 Hz), Alpha (8-12 Hz). We recognize that the highest peak in our spectra reflects a significant difference in the 7-12 Hz range but we wanted to explain the data in terms of the canonical frequency bands in order to make assertions about these previously established frequency windows.

R3.8) Figure 3A; table hard to understand. Explain in caption.

We appreciate the reviewer's attention to this issue as our caption did not contain the detail and explanation to make it sufficiently clear. The caption of Figure 3A has been

expanded with clear details on the breakdown of the table and the plot in 3B. In addition, we further specified column labels in Figure 3A.

Reviewers' Comments:

Reviewer #1 (Remarks to the Author):

The revised manuscript has addressed my concerns only partially.

- 1) The description of Figure 3A at the bottom of p. 2 still is confusing, as the numbers in the figure do not apparently match the numbers in the text related to the number of unique contacts. It thus is very confusing to understand the statistical tests and how they apply to the data that are actually presented in the figure. I assume that the problem is that the numbers in the figure do not correspond to unique contacts, that is, a single contact can contribute to more than one number in the figure if it is significant in more than one comparison. The authors need to provide a table with the unique contacts to support the statistical tests or do a better job of explaining the data behind the tests in the main text.
- 2) Line 103 should say row 7, not row 6
- 3) Line 105: are the numbers in the parentheses correct? delta is 10 vs 3 and alpha is 4 vs. 9 in Fig. 3A, no? If I am incorrect, then this just shows that the text is confusing in how it describes the figure. (I think I understand the figure precisely, just not how it is described in the text.)
- 4) For a future revision, please indicate exactly which lines in a revised manuscript contain revised text, to make the reviewers' job easier in evaluating the revisions.
- 5) I am still not convinced by the authors' arguments about which real world comparison is the best match for the VR task, but since they have made their rationale more explicit, we can let readers decide how much they buy it. The data are still convincing.

Reviewer #2 (Remarks to the Author):

After reading all the responses to reviewers, the revised main manuscript and supplementary information, I believe the authors have adequately addressed all the issues I raised (as well as those raised by the other reviewers) in my initial review. They have provided justification for all the statistical methods and added clarification to the manuscript where needed.

Reviewer #3 (Remarks to the Author):

The authors have adequately addressed the concerns of the referees

R1.1) “The description of Figure 3A at the bottom of p. 2 still is confusing, as the numbers in the figure do not apparently match the numbers in the text related to the number of unique contacts. It thus is very confusing to understand the statistical tests and how they apply to the data that are actually presented in the figure. I assume that the problem is that the numbers in the figure do not correspond to unique contacts, that is, a single contact can contribute to more than one number in the figure if it is significant in more than one comparison. The authors need to provide a table with the unique contacts to support the statistical tests or do a better job of explaining the data behind the tests in the main text.”

We apologize for this confusion. The reviewer is correct that unique contacts were used for statistical tests comparing effects across the 1-12 Hz band because a single electrode could be significant in multiple frequency bands. We have now added a column to the table in Figure 3A which contains the total number of unique electrode contacts across the 1-12Hz band for each contrast. We have added the following explanation to the Figure 3A caption and we have ensured that all references to unique electrode contacts vs. effects within specific frequency bands are now clear.

“Because some contacts showed significant effects in multiple frequency bands (e.g., 1-4 Hz and 4-8 Hz), the final column (“unique 1-12Hz”) tabulates the total number of unique contacts showing effects across the 1-12Hz band for each comparison.”

We are grateful to the reviewer because this has helped clarify the numbers used in our analyses. All results remain unchanged.

R1.2) “Line 103 should say row 7, not row 6”

We thank the reviewer for catching this misstatement – the correction has been made.

R1.3) “Line 105: are the numbers in the parentheses correct? delta is 10 vs 3 and alpha is 4 vs. 9 in Fig. 3A, no? If I am incorrect, then this just shows that the text is confusing in how it describes the figure. (I think I understand the figure precisely, just not how it is described in the text.)”

We thank the reviewer for this comment. We have now fixed this statement to be clearer:

“Using a 2 x 3 Fisher’s exact test to assay for associational differences, we found a significant cross over interaction (Figure 3A row 1 vs. row 7, $p < .04$, Figure 3B). Specifically, this effect appeared driven by a greater number of contacts showing increases in the delta band during virtual than real world searching compared with the alpha band (delta: 10 vs. 3 contacts and alpha: 4 vs. 9 contacts, Figure 3A).”

Reviewers' Comments:

Reviewer #1 (Remarks to the Author):

The authors have addressed my concerns satisfactorily. I have no further concerns with the manuscript.